# `confopt`: A Library for Implementation and Evaluation of Gradient-based One-Shot NAS Methods

Abhash Kumar Jha[1,*]  Shakiba Moradian[1,*]  Arjun Krishnakumar[1,*]  Martin Rapp[3]
Frank Hutter[4,2,1]

[1]Department of Computer Science, University of Freiburg
[2]ELLIS Institute Tübingen
[3]Bosch Center for Artificial Intelligence
[4]Prior Labs
[*]Equal contribution.

**Abstract**  Gradient-based one-shot neural architecture search (NAS) has significantly reduced the cost of exploring architectural spaces with discrete design choices, such as selecting operations within a model. However, the field faces two major challenges. First, evaluations of gradient-based NAS methods heavily rely on the DARTS benchmark, despite the existence of other available benchmarks. This overreliance has led to saturation, with reported improvements often falling within the margin of noise. Second, implementations of gradient-based one-shot NAS methods are fragmented across disparate repositories, complicating fair and reproducible comparisons and further development. In this paper, we introduce Configurable Optimizer (confopt), an extensible library designed to streamline the development and evaluation of gradient-based one-shot NAS methods. Confopt provides a minimal API that makes it easy for users to integrate new search spaces, while also supporting the decomposition of NAS optimizers into their core components. We use this framework to create a suite of new DARTS-based benchmarks, and combine them with a novel evaluation protocol to reveal a critical flaw in how gradient-based one-shot NAS methods are currently assessed. The code can be found under this link.

## 1 Introduction

Neural Architecture Search (NAS), the domain of research that automates the design of neural network architectures, has matured significantly over the past decade. In its early days, most of the methods were based on reinforcement learning (Zoph and Le, 2017; Zoph et al., 2018; Baker et al., 2017; Pham et al., 2018) and evolutionary search methods (Real et al., 2017, 2019) . While these methods were effective, they also demanded substantial computational resources and time. Differentiable Architecture Search (DARTS) (Liu et al., 2019), the formative gradient-based one-shot NAS method, sped up the time required for searching the space by orders of magnitude. Subsequent DARTS-based methods have enabled even more efficient exploration of bounded architectural spaces, while also addressing several of the challenges of DARTS (White et al., 2023).

A major challenge in gradient-based one-shot NAS lies in reliable benchmarking and evaluation. Prior work has shown that the performance of a given architecture is heavily influenced by the training recipe, and even the random *seeds* used, making fair comparisons between NAS methods difficult (Yang et al., 2020). Despite this, the DARTS search space remains the primary benchmark for evaluating new gradient-based NAS approaches. Furthermore, many of the reported improvements from newer methods fall within the margin of noise, making it challenging to draw confident conclusions (Zhang and Ding, 2023).

In this work, we highlight several challenges in evaluating NAS methods that rely on a supernet, as done in DARTS, and propose a new evaluation protocol designed to address these issues. To

support this effort, we introduce *Configurable Optimizer* (*confopt*), a library built specifically for the development and evaluation of gradient-based one-shot NAS methods. Using this library, we construct **DARTS-Bench-Suite**, a collection of benchmarks derived from the DARTS search space, and use them to demonstrate key flaws in the prevailing evaluation pipeline. Each benchmark is an instantiation of the DARTS search space with a distinct configuration, varying in candidate operations, network depth, width, and other architectural details. Importantly, while each benchmark retains a large and expressive search space, the associated supernets are significantly more efficient to train. Concretely, we summarize our contributions as follows:

1. We introduce *Configurable Optimizer (confopt)*, a library for developing and benchmarking gradient-based one-shot NAS methods.
2. We introduce DARTS-Bench-Suite, a benchmark suite of nine DARTS-based benchmarks to more comprehensively evaluate NAS methods.
3. We evaluate seven NAS optimizers across nine new benchmarks and find that their rankings differ substantially across these settings, highlighting the need for a more comprehensive evaluation of gradient-based one-shot NAS methods.

## 2 Background and Related Work

We begin by reviewing a few important prior works in the field of gradient-based one-shot NAS.

### 2.1 DARTS

DARTS (Liu et al., 2019) introduced a seminal reformulation of the NAS problem by transforming the discrete architecture search into a continuous optimization problem. It defines a search space where the network architectures are represented as directed acyclic graphs, with each node in the graph as a feature map and each edge as a candidate operation. It relaxes the space of candidate operations by expressing each edge as a weighted sum of all possible operations. For an input feature $x$ to incoming node $i$ of an edge $(i, j)$ with an operation set $\mathcal{O}$, the feature $\bar{o}_{i,j}$ at output node $j$ is a continuous relaxation of operations $o \in \mathcal{O}$.

$$\bar{o}_{i,j}(x) = \sum_{o \in \mathcal{O}} \frac{\exp(\alpha_o^{(i,j)})}{\sum_{o' \in \mathcal{O}} \exp(\alpha_{o'}^{(i,j)})} o(x),$$

where $\alpha_o^{(i,j)}$ is the architectural parameter for the edge $(i, j)$ indicating the strength of the candidate operation $o$.

Formally, DARTS can be stated as a bi-level optimization problem such that the outer level optimizes architectural parameters $\alpha$ by minimizing the validation loss $\mathcal{L}_{val}$, constraining the inner level to optimize the network weights $w$ by minimizing the training loss $\mathcal{L}_{train}$. At the end of the optimization, DARTS performs the discretization of the architectural parameter $\alpha$ to obtain an architecture by selecting the operation having the highest architectural weight for each edge.

$$\min_{\alpha} \; \mathcal{L}_{val}(w^*(\alpha), \alpha) \qquad \text{such that,} \quad w^*(\alpha) = \arg\min_{w} \; \mathcal{L}_{train}(w, \alpha)$$

DARTS and related methods search for an optimal *cell* within a *supernet*, which is described by the learned architectural parameters. The supernet is a large network encompassing all possible architectures within the search space. Its *macro-architecture* is composed of multiple stacked cells.

### 2.2 Evaluation Protocol

Once an optimal cell architecture is learned during the supernet training phase, a target model is constructed by stacking this cell multiple times. This model is then trained from scratch and evaluated on a hold-out test dataset. Notably, the size of the supernet does not always match the

target model. The target model can incorporate a greater number of stacked cells or operate with a larger set of channels. Moreover, the dataset used for training the supernet is the same one employed for training the derived discrete model. To account for variability introduced by random seeds in the training pipeline, it is also common practice to train the target model multiple times and report the mean and variance of its test accuracy.

### 2.3 Related Works

While efficient, DARTS suffers from high memory consumption and instability in architecture selection, which limit its scalability. Several studies (Cai et al., 2019; Chen et al., 2019) address these issues through distinct optimizations. For instance, GDAS (Dong and Yang, 2019) mitigates memory overhead by hard-sampling sub-graphs using Gumbel-Softmax, thereby reducing conflicts in gradient updates. Similarly, PC-DARTS (Xu et al., 2019) decreases memory consumption by sampling a small portion of channels, enhancing stability. Beyond resource constraints, DARTS also experiences performance collapse due to skip connection dominance and discretization gaps. Some works alleviate this problem by directly addressing overfitting through loss landscape metrics to track optimization (Zela et al., 2020; Jiang et al., 2023; Movahedi et al., 2023), introducing perturbations (Chen and Hsieh, 2020), or reframing the problem as a distribution learning task (Chen et al., 2021). These approaches target orthogonal aspects such as efficiency, scalability, and robustness, collectively improving DARTS' applicability and underscoring the need for a unified codebase that streamlines these optimizations.

### 2.4 Benchmarks

Recent efforts have sought to unify and systematically evaluate Neural Architecture Search (NAS) optimizers, aiming to establish rigorous comparative benchmarks. NAS-Bench-360 (Tu et al., 2022) expanded NAS evaluation beyond conventional image classification by incorporating diverse real-world tasks, assessing three distinct search spaces. Similarly, NAS-Bench-Suite (Mehta et al., 2022) provides a comprehensive analysis of NAS algorithms across combinations of search spaces and datasets, demonstrating a persistent lack of generalization across existing benchmarks. This work leverages tabular benchmarks such as NAS-Bench-101 (Ying et al., 2019), NAS-Bench-201 (Dong and Yang, 2020), and surrogate models like NAS-Bench-301 (Zela et al., 2022) to rank NAS methods. Complementary libraries—including Microsoft's Archai (Shah et al., 2020), aw_nas (Ning et al., 2021), and DeepArchitect (Negrinho and Gordon, 2017; Negrinho et al., 2019)—offer platforms for implementing AutoML algorithms but often lack standardized evaluation protocols.

Closely related to our contribution, Zhang et al. (2023) introduced the Large and Harder DARTS Search Space (LHD), a challenging benchmark designed to evaluate transductive robustness under varied discretization policies. While their work exposes critical limitations in DARTS-based evaluations, our study extends this analysis through a unified codebase that systematically investigates variations of the DARTS search space.

## 3 Why A New Evaluation Protocol?

In this section, we motivate the need for upgrading the evaluation protocol of gradient-based one-shot NAS methods.

### 3.1 Points of Failure in the DARTS Pipeline

DARTS trains a *proxy* supernet to search for the optimal *cell* architecture—"proxy" because this supernet has fewer cells and fewer channels than the *target* model. This cell is then stacked multiple times, with more channels, to obtain the target model. There are two points of failure in this sequence. 1) The method might discover sub-optimal cell architectures for the given proxy supernet. 2) The optimal architecture for the proxy supernet might be suboptimal for the target network. In principle, all methods that search a proxy supernet would be affected by this.

**Sub-optimal architectures**: The first issue is particularly evident in DARTS, where the supernet training phase suffers from a *discretization gap*—a phenomenon in which the trained supernet performs well, but the discretized architecture derived from it performs poorly, largely due to the excessive presence of parameterless skip connections. Subsequent methods have focused on improving this, with several works highlighting the role of the Hessian eigenvalues of the model parameters in predicting model performance (Zela et al., 2020; Chen and Hsieh, 2020).

**Poor rank correlation of proxy model**: This challenge pertains to the low rank correlation between models on a *proxy* space (i.e., architectures with fewer cells or channels) and the *target* space, which are typically much larger. One direct approach for mitigating poor rank correlation is to eliminate the proxy supernet altogether and train a full-scale supernet that directly matches the target model. However, this was infeasible due to GPU memory constraints. This prompted a line of research into memory-efficient gradient-based one-shot NAS methods. While some of these methods were compatible with existing methods, such as P-DARTS (Chen et al., 2019), some others, such as ProxylessNAS (Cai et al., 2019), introduced entirely new optimization schemes to improve scalability. Recent advancements in hardware—particularly the availability of AI/ML-specialized GPUs and better support for distributed training across multiple GPUs–now make fitting larger supernets more feasible. We argue that the rank correlation between the proxy supernet and the discrete model should be factored out when evaluating the NAS method.

## 3.2 Desiderata for NAS Evaluation

Ideally, a NAS method should be able to identify optimal architectures across any given search space. To properly assess this capability, the NAS evaluation pipeline must be as rigorous as possible. Achieving this requires that the evaluation protocol satisfy two key desiderata.

### 3.2.1 Evaluation on multiple search spaces.
The DARTS and NAS-Bench-201 search spaces are among the most widely used in the literature on gradient-based one-shot NAS, with the former serving as the primary benchmark for evaluating new methods. A direct consequence of the nature of the DARTS search space has been a substantial body of work focused on addressing challenges associated with this particular search space (Chu et al., 2020, 2021; Li et al., 2021; Bi et al., 2019; Wang et al., 2021; Zela et al., 2020; Chen and Hsieh, 2020; Jiang et al., 2023; Chu et al., 2021). To avoid that new NAS methods *overfit* their recipes to the challenges of the DARTS search space, they must be evaluated on other similarly difficult benchmarks as well.

### 3.2.2 Unbiased evaluation of architecture performance.
To the extent that it is feasible, the evaluation pipeline should assess the *intrinsic* quality of the architectures discovered by a NAS method. This requires satisfying two key conditions. 1) The discovered architectures should generalize well to the data distribution on which the supernet was trained, and the evaluation protocol should be designed to ensure this. 2) The evaluation should isolate the intrinsic quality of the discovered architecture, minimizing confounding factors such as hyperparameter selection.

**Generalization**: Traditionally, one-shot NAS methods are evaluated in two stages: the supernet is first trained on a dataset, and then the derived discrete architecture is re-trained from scratch on the same training samples. This setup complicates the assessment of true generalization, as it becomes unclear whether performance is due to the architecture's intrinsic quality or prior exposure to the training data. To further examine generalization, it is common practice to transfer the discovered architecture to a dataset such as ImageNet (Zoph et al., 2018; Liu et al., 2018, 2019), which has a substantially different data distribution. While this assesses cross-dataset generalization, it does not directly evaluate how well the architecture generalizes on the original distribution.

**Minimize confounding factors**: The argument that even the *seeds* of evaluation pipelines can induce variation in performance of the models has been made before (Yang et al., 2020) but the same consideration is seldom given to their hyperparameters. When architectures are re-trained using a fixed set of hyperparameters, those that are naturally better aligned with this specific

configuration may appear superior—not necessarily due to their structures, but due to incidental compatibility—thus introducing bias in the evaluation.

### 3.2.3 Cheap Evaluation

. The evaluation of models by training them from scratch is still time-consuming. A single trial of a DARTS model takes approximately 18 to 36 hours to train on an Nvidia GeForce RTX 2080. The computational cost of training models from scratch significantly limits the ability of NAS researchers to evaluate methods across multiple search spaces. Ideally, this process should be more efficient to enable broader and more comprehensive evaluations.

## 4 The Configurable Optimizer Library

In this section, we discuss *Configurable Optimizer*, the library designed for developing and evaluating gradient-based one-shot NAS methods.

### 4.1 Design of the library

1. **Architecture samplers**. These constitute the strategy used by a method to sample the architectural parameters. DARTS, for example, represents the architectural parameters as a set of parameters in a continuous space, with no sampling strategy. However, more advanced methods, such as DrNAS or GDAS, sample from a distribution that is defined by the architectural parameters.

2. **Modifications to the supernet**. NAS methods might modify the operations of the search space. Consider, for example, PC-DARTS, which learns only *partial channels* of the convolutional operations at once. This involves modifying the operations in the supernet. In a similar vein, NAS methods might employ *weight-entanglement* instead of *weight-sharing* in the candidate operations (Sukthanker et al., 2024), or use low rank decompositions to train them (Krishnakumar et al., 2024).

3. **Regularization terms**. Several methods add penalty terms to the validation loss as part of the method. DrNAS and FairDARTS are two prominent examples.

4. **Pruning operations**. Methods such as DrNAS prune operations at specific epochs, thereby eliminating them from the set of possible optimal operations for that edge. Pruning speeds up the supernet training too as fewer operations have to be learned in every step.

5. **Early stopping techniques**. Methods such as DARTS+ (Liang et al., 2019) and RobustDARTS (Zela et al., 2020) employ early stopping of the training of the supernet. The early stopping may be based on heuristics encoded into the algorithm, as in DARTS+, which terminates supernet training if the number of skip connections in the dominant architecture after a training epoch exceeds a certain threshold.

Configurable Optimizer (confopt) distinguishes between two aspects of a NAS method: algorithmic components—such as strategies for sampling architectures or regularizing the loss—and modifications to the supernet itself, such as employing *partial connections* or *weight entanglement* within convolutional candidate operations, or pruning the candidate operations. To streamline this interaction, supernet mutations are implemented independently of the NAS methods that utilize them. For instance, partial connections are defined directly at the supernet level. When combined with a DARTS sampler and edge normalization, this configuration yields PC-DARTS.

To make it easier and more intuitive for users to extend the library with new search spaces (i.e., supernets), confopt follows two core design principles:

1. **A minimal set of core APIs**. Supernet classes in confopt act as wrappers around existing supernet implementations. These wrappers enforce a consistent interface by requiring only a small set

of core APIs. This design offers two major benefits. First, integrating a new search space into confopt does not require re-implementing it from scratch. Users can simply add their existing code to the repository and implement a few wrapper APIs to bridge it. Second, this abstraction enables the confopt training loop to operate independently of the specific search space, treating all supernets in a uniform way.

2. **Optional APIs.** While the core APIs ensure minimal integration overhead, users can implement additional optional APIs to unlock more advanced functionality. For example, they may expose metrics to track during training, such as gradient statistics of architectural parameters, gradient matching scores, or layer alignment scores.

The library uses *Profile* classes to fully describe the NAS method that is to be run on a given search space. It can be used to configure everything about the supernet training phase, from the settings of the supernet (including mutations to be made to it), the type of architecture samplers, to the seeds to use in the experiment. Confopt already provides default profiles for existing methods like DARTS, DrNAS or GDAS. The user may also choose to instantiate a custom Profile. Lastly, the library offers seamless integration with Weights and Biases (Biewald, 2020), allowing the user to track several important metrics as the supernet training progresses.

## 4.2 Code Example

A minimal code example of running GDAS on the DARTS search space is shown in Listing 1. The `GDASProfile` is pre-configured with all the default components for the GDAS method. The `trainer_config` argument being set to `SearchSpaceType.DARTS` tells the `Profile` class to use the default DARTS training configuration while training the supernet. The search space to use is specified in the instantiation of the `Experiment` class, along with the dataset and the seed to set before training. A more advanced example is shown in Listing 2 of Appendix C.

```python
from confopt.profile import GDASProfile
from confopt.train import Experiment
from confopt.enums import SearchSpaceType, DatasetType

profile = GDASProfile(trainer_config=SearchSpaceType.DARTS, epochs=50)
experiment = Experiment(
                search_space=SearchSpaceType.DARTS,
                dataset=DatasetType.CIFAR10,
                seed=9001
            )
experiment.train_supernet(profile)
```

Listing 1: Example of minimal code to run GDAS on the DARTS search space

## 5 The DARTS-Bench-Suite

We design the DARTS-Bench-Suite to address the challenges mentioned in Section 3. We address the challenge discussed in Section 3.2.1 by introducing nine benchmarks derived from the DARTS search space, combining three supernet architecture variants with three sets of candidate operations. The supernet variants were designed to maintain approximately the same number of learnable parameters, around 1M. They are as follows:

1. *DARTS-Wide* allows the supernet to have more initial channels than the original DARTS supernet but limits its depth, i.e., the number of cells.
2. *DARTS-Deep* imposes a lower channel count while increasing depth.

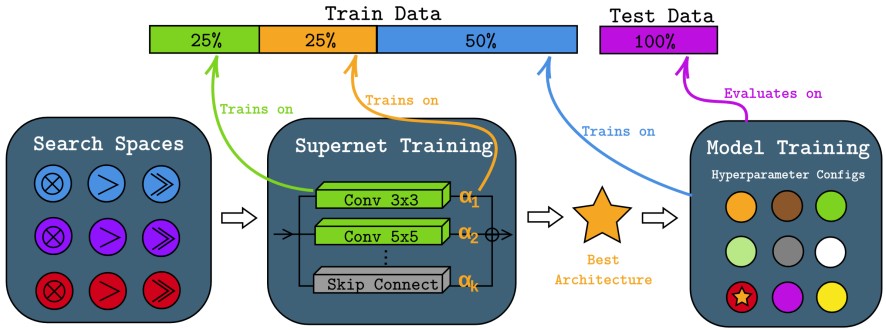

Figure 1: An overview of DARTS-Bench-Suite. The benchsuite consists of nine benchmarks, which are a combination of three supernet variants with three sets of candidate operations. The NAS method is run on each benchmark thrice. Then, the architecture from the trial with the lowest validation loss is chosen to be evaluated for each benchmark. A discrete model with this architecture is then trained on nine different hyperparameter configurations from scratch, and the average and best test accuracies are reported. Note that supernet and discrete model training both use disjoint sets of training data.

3. *DARTS-Single-Cell* consists of only a single cell in the macro architecture but doubles the number of *intermediate* nodes in the cell, increasing the number of learnable edges from 14 to 44. This leads to an exponential growth in the number of possible architectures within the search space. Furthermore, DARTS-Single-Cell has a higher number of initial channels.

Table 1 highlights the differences between the search spaces. We introduce the following three sets of candidate operations:

1. *Regular* contains the same operations as DARTS.
2. *No-skip* contains the same operations as DARTS, sans skip connection.
3. *All-skip* has the same operations as DARTS, but with an auxiliary skip connection for all parametric operations.

Further, we make the target network match the size of the supernet. This has two advantages. First, it factors out rank correlation from the assessment of the NAS method, addressing the issue discussed in Section 3.1. Second, it makes the target models significantly smaller. Considering that they are now trained on half as much data as well, the training of the discrete models becomes much faster, thereby addressing the expensive evaluation of the methods (Section 3.2.3).

## 5.1 Evaluation

To address the challenges mentioned in Section 3.2.2, we propose a novel evaluation protocol for one-shot NAS methods. **First**, we split the training dataset into two halves: the first half is used to train the supernet, while the second half is used to train the discrete models. This ensures that the final architecture is evaluated on unseen data drawn from the same distribution. **Second**, we train each discrete model obtained from the supernet search using nine different hyperparameter configurations. Then we report both the best and average performance across these runs. This approach provides a more unbiased assessment of an architecture's intrinsic quality, independent of hyperparameter tuning. These hyperparameters are not chosen using hyperparameter optimization (HPO). They constitute a $3 \times 3$ grid of *learning rate* and *weight decay*, obtained as a cross product of perturbations of the original DARTS hyperparameters. See Table 15 of Appendix B.2 for the exact values. A high-level overview of DARTS-Bench-Suite is given in Figure 1.

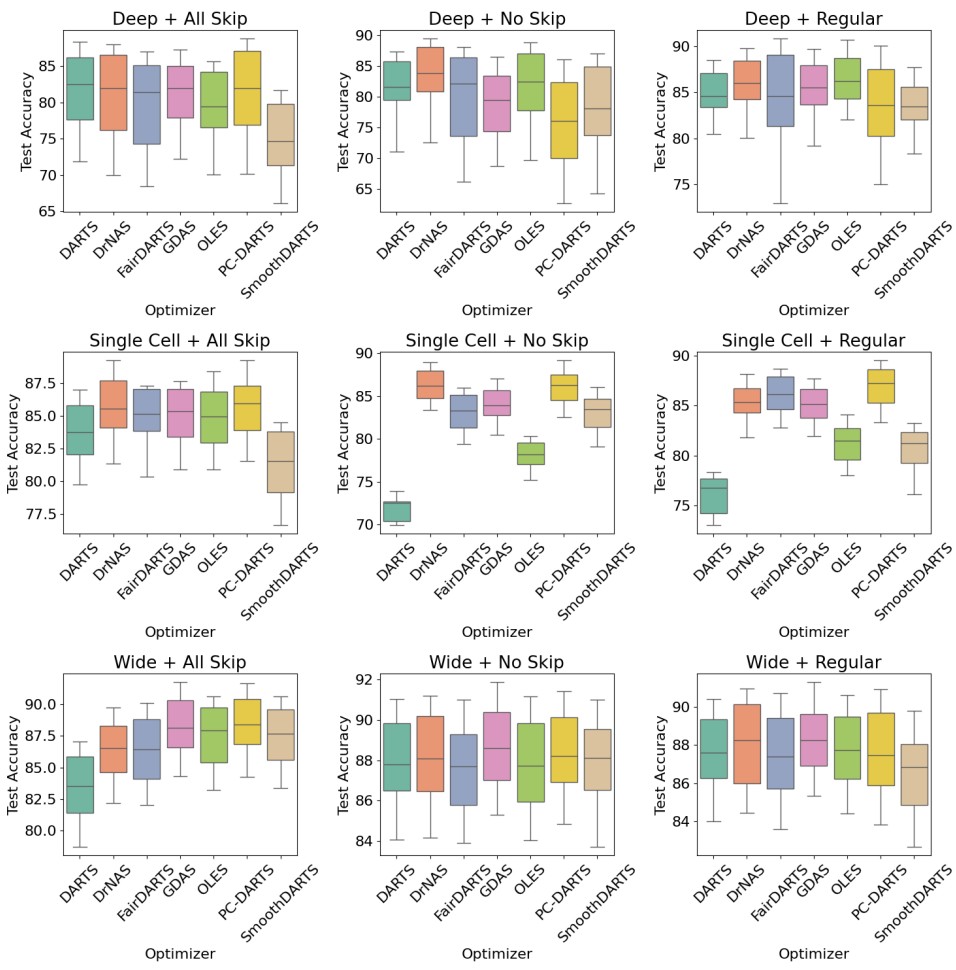

Figure 2: Performance of the NAS methods on all nine benchmarks across nine evaluations.

## 6 Experiments

In this section, we assess seven NAS methods on the nine search spaces of DARTS-Bench-Suite. The methods are as follows: DARTS, PC-DARTS, FairDARTS, SmoothDARTS, OLES, DrNAS, and GDAS. We first train the supernet using the NAS method with three fixed seeds on the supernet training split. From the resulting candidates, we select the architecture with the lowest validation loss. This selected architecture is then trained from scratch using nine distinct hyperparameter configurations on the model training split. Finally, all resulting models are evaluated on the test set. The results across the nine benchmarks are visualized in Figure 2. They can also be found in Tables 2-10. We now take a closer look at this data to answer two questions.

### 6.1 Does the choice of hyperparameters to train the model matter for the rank of NAS methods?

To answer this question, we observe the rankings of the seven NAS methods we evaluated using three different hyperparameter configurations. First, using the default hyperparameter configuration used by DARTS. Then, with a fixed configuration from among the nine. Finally, we use whichever hyperparameter configuration yielded the *best* test accuracy.

The results are shown in Figure 3. Clearly, the choice of hyperparameters makes a difference. We note that in the 63 architectures which were each trained with nine different hyperparameter configurations (HP1-9), HP6 was the best 53 times, HP5 eight times, and HP4 twice. These three

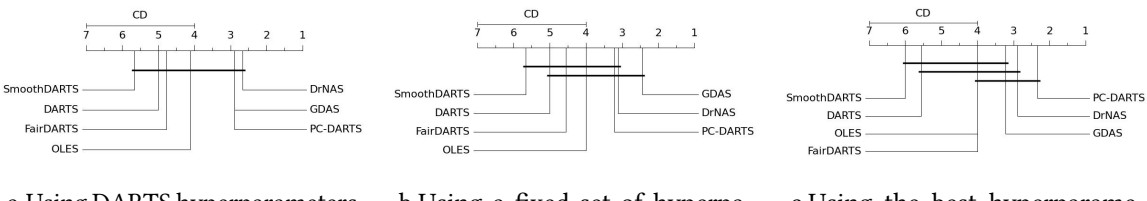

a Using DARTS hyperparameters (HP2)

b Using a fixed set of hyperparameters (HP8)

c Using the best hyperparameters

Figure 3: CD Plots of the NAS Methods

configurations had a learning rate of 0.1 in common. We did not tune the hyperparameter configurations a priori. This result suggests that doing so could be good practice—perhaps sampling a few architectures and training them with different hyperparameters to understand the *neighborhood* of good configurations could be useful in selecting the hyperparameter sets used to train and evaluate the discrete models.

## 6.2 Are the rankings of NAS methods stable across the different benchmarks?

Figure 4 in Appendix A shows the rank correlation between the nine benchmarks in DARTS-Bench-Suite on the seven methods that we evaluated. The win rate of the methods against each other is shown in Figure 6 of Appendix A. The lack of clear ranking across benchmarks is interesting, especially considering that they are variants of the same underlying search space, i.e., DARTS. This is particularly pronounced between the *Wide* and the *Deep* benchmarks, with the rankings being moderately *anti-correlated* between *Wide + All Skip* and *Deep + No Skip*. **The rank disparity, even with confounding factors removed and the influence of hyperparameters accounted for, indicates that the evaluation of the methods on only the DARTS search space is brittle**. It highlights the need for more comprehensive evaluation across different search spaces to ascertain the superiority of one method over another—evaluation on the DARTS search space alone is insufficient.

## 7 Conclusion

With this work, we present Configurable Optimizer (confopt), a library specifically designed for implementing and comparing gradient-based one-shot NAS methods. We use it to develop DARTS-Bench-Suite, a collection of nine DARTS-based benchmarks. Using DARTS-Bench-Suite, we show that the relative performances of NAS methods are inconsistent when changes are applied to the search space. The current practice of strongly relying on the original DARTS search space to rank NAS methods can be misleading. We call on the community to design more robust search spaces for evaluating NAS methods, as well as to develop novel methods that achieve statistically significant improvements. We hope that *confopt* will serve as a useful tool in advancing this effort.

    **Current limitations of confopt and DARTS-Bench-Suite**: Despite introducing more heterogeneous search spaces, we still rely heavily on the DARTS search space. However, confopt allows users to easily integrate new search spaces. Similarly, we currently use standard datasets like CIFAR. More generalizable findings could be obtained by introducing more diverse, real-world datasets.

### Acknowledgments and Disclosure of Funding

Frank Hutter acknowledges the financial support of the Hector Foundation. Robert Bosch GmbH is acknowledged for financial support. This research was partially supported by TAILOR, a project funded by EU Horizon 2020 research and innovation programme under GA No 952215.

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

# A  Full Results

This section summarizes the complete results of the experiments. Figure 4 shows the correlation between the rankings of the seven NAS methods across all nine benchmarks. For a given benchmark, the NAS methods are ranked by considering the *best* test performance across all nine hyperparameter configurations (HP1-HP9) for each method. Figure 6 shows the win rates for the same. Similarly, Figure 5 shows the correlation between the benchmarks, but using the *mean* test accuracy of the nine hyperparameter configurations for each method. Figure 7 highlights the win rates for this approach.

Tables 2 to 10 show the mean and maximum accuracies of the architectures discovered by the NAS methods for each benchmark. Tables 11 and 12 summarize the rankings of the optimizers across the different *operation sets* (*Regular, No-Skip, All-Skip*) and *supernet variants* (*DARTS-Wide, DARTS-Deep, DARTS-Single-Cell*), respectively. The rankings of the NAS methods for each of the benchmarks are shown in Table 13.

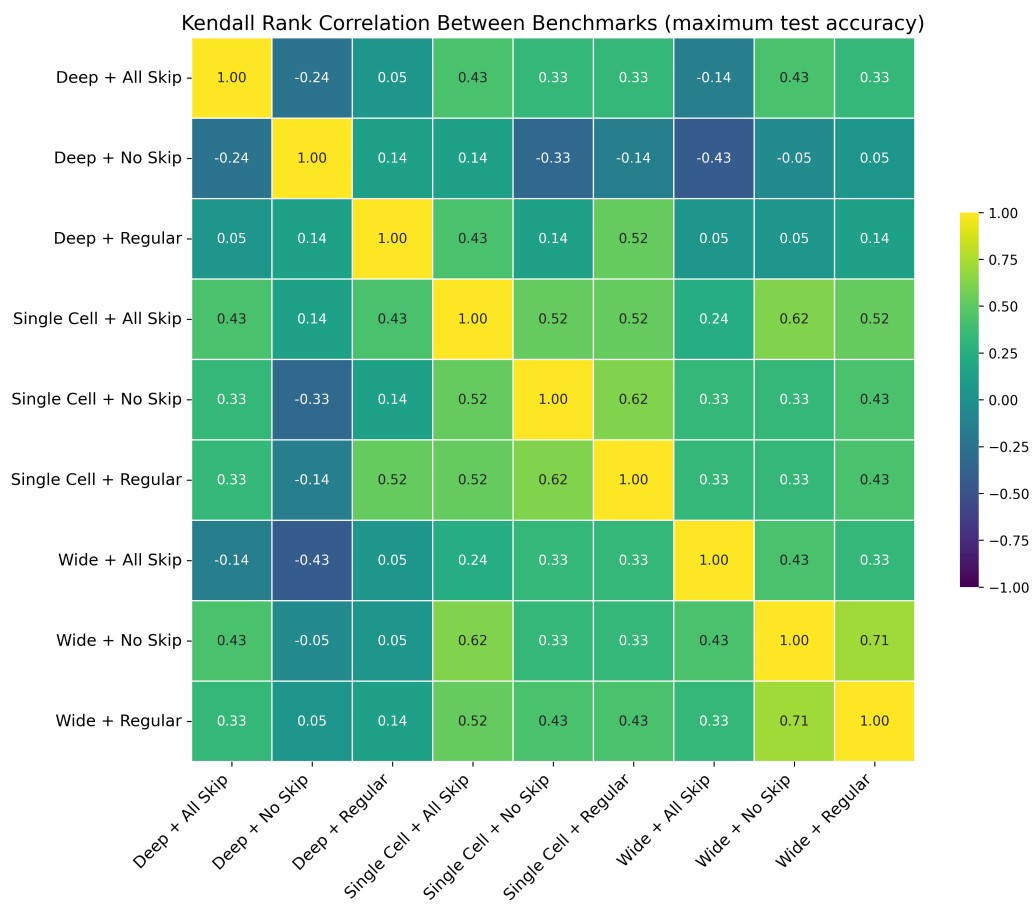

Figure 4: Correlation between benchmarks (based on the *best* test accuracy obtained per NAS method)

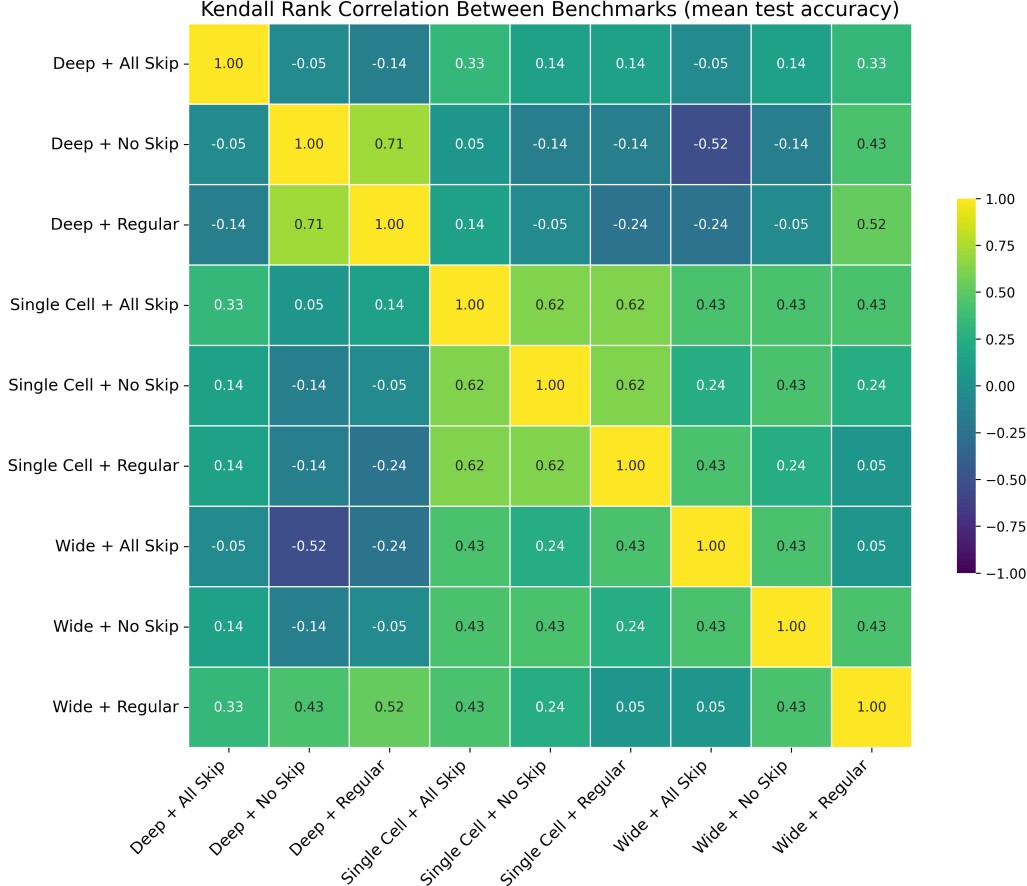

Figure 5: Correlation between benchmarks (based on the *mean* test accuracy per NAS method)

| Search Space | Cells | Initial Channels | Edges per cell |
|---|---|---|---|
| DARTS | 8 | 16 | 14 |
| DARTS-Wide | 4 | 18 | 14 |
| DARTS-Deep | 16 | 7 | 14 |
| DARTS-Single-Cell | 1 | 26 | 44 |

Table 1: Search space configurations.

| Method | Mean Test Accuracy (%) | Maximum Test Accuracy (%) |
|---|---|---|
| DARTS | 84.92 ± 2.90 | 88.49 |
| GDAS | 85.30 ± 3.70 | 89.68 |
| PC-DARTS | 83.15 ± 5.52 | 90.03 |
| Fair DARTS | 83.73 ± 6.59 | **90.85** |
| SmoothDARTS | 83.37 ± 3.42 | 87.73 |
| DrNAS | 85.75 ± 3.28 | 89.75 |
| OLES | **86.45 ± 2.99** | 90.71 |

Table 2: Test Accuracy for Deep x Regular

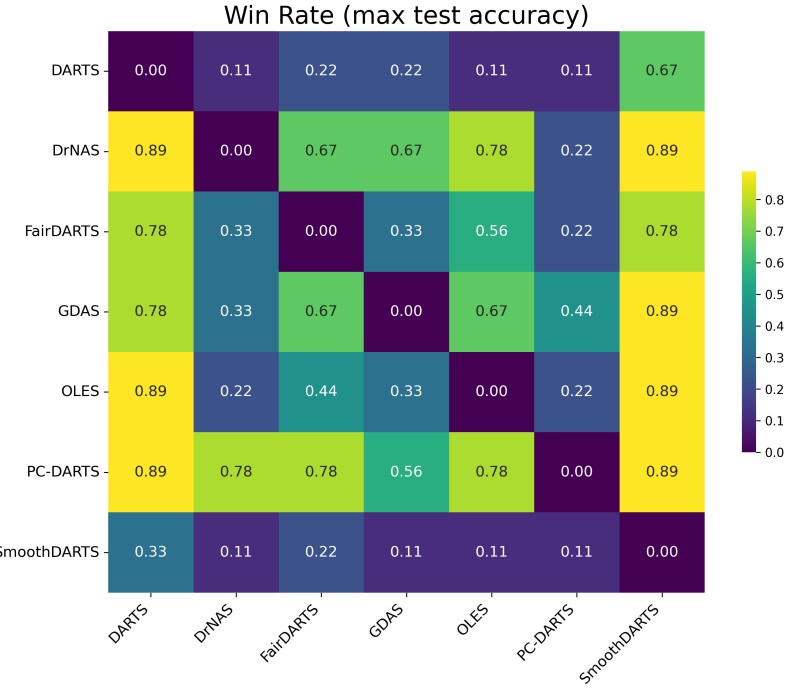

Figure 6: Win rates of NAS methods (*max* test accuracy)

| Method | Mean Test Accuracy (%) | Maximum Test Accuracy (%) |
|---|---|---|
| DARTS | 81.19 ± 6.03 | 87.35 |
| GDAS | 78.48 ± 6.63 | 86.54 |
| PC-DARTS | 75.37 ± 9.16 | 86.02 |
| Fair DARTS | 79.19 ± 8.80 | 88.15 |
| SmoothDARTS | 77.87 ± 8.74 | 87.06 |
| DrNAS | **83.04 ± 6.36** | **89.49** |
| OLES | 81.25 ± 7.23 | 88.85 |

Table 3: Test Accuracy for Deep x No Skip

| Method | Mean Test Accuracy (%) | Maximum Test Accuracy (%) |
|---|---|---|
| DARTS | **81.44 ± 6.21** | 88.34 |
| GDAS | 81.03 ± 5.38 | 87.33 |
| PC-DARTS | 81.38 ± 6.86 | **88.84** |
| Fair DARTS | 79.56 ± 7.02 | 87.05 |
| SmoothDARTS | 74.96 ± 5.86 | 81.72 |
| DrNAS | 80.86 ± 6.79 | 88.02 |
| OLES | 79.16 ± 6.17 | 85.70 |

Table 4: Test Accuracy for Deep x All Skip

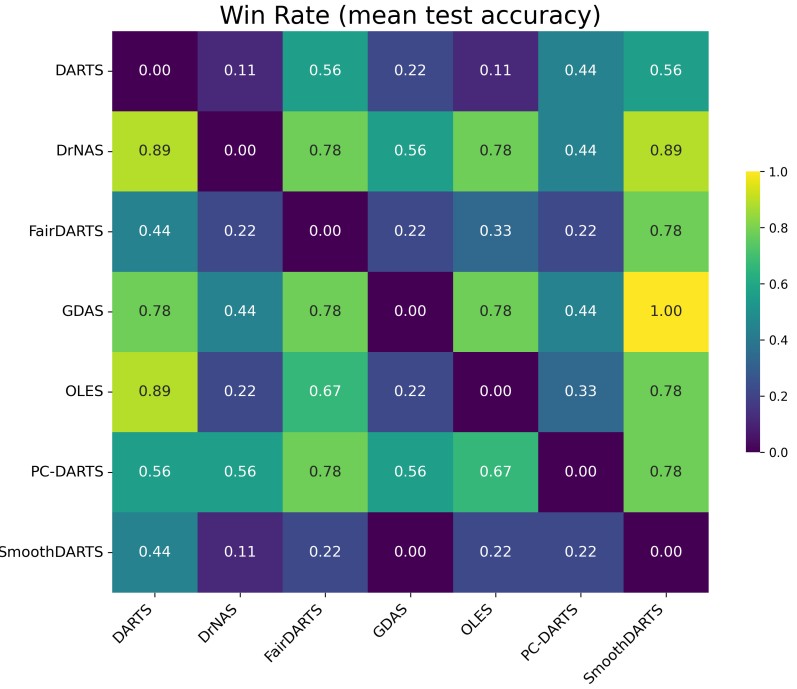

Figure 7: Win rates of NAS methods (*mean* test accuracy)

| Method | Mean Test Accuracy (%) | Maximum Test Accuracy (%) |
|---|---|---|
| DARTS | 87.63 ± 2.31 | 90.41 |
| GDAS | **88.30 ± 2.16** | **91.30** |
| PC-DARTS | 87.57 ± 2.53 | 90.93 |
| Fair DARTS | 87.35 ± 2.53 | 90.73 |
| SmoothDARTS | 86.51 ± 2.48 | 89.80 |
| DrNAS | 87.91 ± 2.50 | 90.94 |
| OLES | 87.77 ± 2.31 | 90.63 |

Table 5: Test Accuracy for Wide x Regular

| Method | Mean Test Accuracy (%) | Maximum Test Accuracy (%) |
|---|---|---|
| DARTS | 87.74 ± 2.52 | 91.05 |
| GDAS | **88.69 ± 2.38** | **91.88** |
| PC-DARTS | 88.33 ± 2.43 | 91.44 |
| Fair DARTS | 87.61 ± 2.53 | 91.01 |
| SmoothDARTS | 87.85 ± 2.50 | 90.99 |
| DrNAS | 88.20 ± 2.53 | 91.20 |
| OLES | 87.81 ± 2.68 | 91.16 |

Table 6: Test Accuracy for Wide x No Skip

| Method | Mean Test Accuracy (%) | Maximum Test Accuracy (%) |
| --- | --- | --- |
| DARTS | 83.54 ± 3.07 | 87.02 |
| GDAS | 88.35 ± 2.65 | **91.76** |
| PC-DARTS | **88.39 ± 2.71** | 91.66 |
| Fair DARTS | 86.32 ± 3.08 | 90.08 |
| SmoothDARTS | 87.47 ± 2.80 | 90.60 |
| DrNAS | 86.31 ± 2.84 | 89.76 |
| OLES | 87.51 ± 2.81 | 90.61 |

Table 7: Test Accuracy for Wide x All Skip

| Method | Mean Test Accuracy (%) | Maximum Test Accuracy (%) |
| --- | --- | --- |
| DARTS | 76.04 ± 2.05 | 78.35 |
| GDAS | 85.18 ± 2.13 | 87.72 |
| PC-DARTS | **86.84 ± 2.36** | **89.54** |
| Fair DARTS | 86.08 ± 2.21 | 88.68 |
| SmoothDARTS | 80.62 ± 2.53 | 83.25 |
| DrNAS | 85.20 ± 2.14 | 88.16 |
| OLES | 81.20 ± 2.19 | 84.09 |

Table 8: Test Accuracy for Single Cell x Regular

| Method | Mean Test Accuracy (%) | Maximum Test Accuracy (%) |
| --- | --- | --- |
| DARTS | 72 ± 1.45 | 73.9 |
| GDAS | 84.03 ± 2.22 | 87.01 |
| PC-DARTS | 85.90 ± 2.37 | **89.16** |
| Fair DARTS | 83.19 ± 2.47 | 85.98 |
| SmoothDARTS | 82.98 ± 2.51 | 86.01 |
| DrNAS | **86.39 ± 1.98** | 88.95 |
| OLES | 78.10 ± 1.87 | 80.30 |

Table 9: Test Accuracy for Single Cell x No Skip

| Method | Mean Test Accuracy (%) | Maximum Test Accuracy (%) |
| --- | --- | --- |
| DARTS | 83.74 ± 2.67 | 87.00 |
| GDAS | 84.89 ± 2.49 | 87.63 |
| PC-DARTS | **85.60 ± 2.78** | **89.25** |
| Fair DARTS | 84.76 ± 2.50 | 87.28 |
| SmoothDARTS | 81.27 ± 2.83 | 84.50 |
| DrNAS | 85.58 ± 2.82 | 89.23 |
| OLES | 84.83 ± 2.77 | 88.42 |

Table 10: Test Accuracy for Single Cell x All Skip

| Optimizer Benchmark | DARTS | DrNAS | Fair DARTS | GDAS | OLES | PC-DARTS | SmoothDARTS |
|---|---|---|---|---|---|---|---|
| Deep + All Skip | **1** | 4 | 5 | 3 | 6 | 2 | 7 |
| Deep + No Skip | 3 | **1** | 4 | 5 | 2 | 7 | 6 |
| Deep + Regular | 4 | 2 | 5 | 3 | **1** | 7 | 6 |
| Single Cell + All Skip | 6 | 2 | 5 | 3 | 4 | **1** | 7 |
| Single Cell + No Skip | 7 | **1** | 4 | 3 | 6 | 2 | 5 |
| Single Cell + Regular | 7 | 3 | 2 | 4 | 5 | **1** | 6 |
| Wide + All Skip | 7 | 6 | 5 | 2 | 3 | **1** | 4 |
| Wide + No Skip | 6 | 3 | 7 | **1** | 5 | 2 | 4 |
| Wide + Regular | 4 | 2 | 6 | **1** | 3 | 5 | 7 |

Table 13: Ranking of NAS methods on the DARTS-Bench-Suite.

| Optimizer Opset | DARTS | DrNAS | Fair DARTS | GDAS | OLES | PC-DARTS | SmoothDARTS |
|---|---|---|---|---|---|---|---|
| All-skip | 6 | 3 | 5 | 2 | 4 | **1** | 7 |
| No-skip | 7 | **1** | 3 | 2 | 6 | 4 | 5 |
| Regular | 7 | **1** | 4 | 2 | 5 | 3 | 6 |

Table 11: Ranking of methods across operation sets.

| Optimizer Subspace | DARTS | DrNAS | Fair DARTS | GDAS | OLES | PC-DARTS | SmoothDARTS |
|---|---|---|---|---|---|---|---|
| Deep | 2 | **1** | 5 | 4 | 3 | 6 | 7 |
| Single-cell | 7 | 2 | 4 | 3 | 6 | **1** | 5 |
| Wide | 7 | 4 | 6 | **1** | 3 | 2 | 5 |

Table 12: Ranking of methods across supernet variants.

## B  Experiment Details

### B.1  Architecture Search

- **Methods**: We evaluate neural architecture search (NAS) methods using three primary samplers: DARTS (Liu et al., 2019), DrNAS (Chen et al., 2021), and GDAS (Dong and Yang, 2019), alongside SDARTS (Chen and Hsieh, 2020), FairDARTS (Chu et al., 2020), PC-DARTS (Xu et al., 2019), and OLES  (Jiang et al., 2023) (adopting DARTS' configurations).

- **Training Protocols**: Liu et al. (2019) conduct searches for 50 epochs; however, failure modes typically emerge beyond this point. To more effectively capture these failure modes, we expand our benchmark to cover a wider range of conditions. Specifically, we run DARTS-based methods for 100 epochs, while DrNAS and GDAS are trained for 100 and 300 epochs, respectively. For PC-DARTS and DrNAS, we implement partial channel sampling with $K = 4$ and a 15-epoch warm-up phase (architecture parameters frozen) to stabilize optimization. FairDARTS and DrNAS apply $L_1$ ($\lambda = 10$) and $L_2$ ($\lambda = 1$) regularization, respectively. Note that, we omit the progressive pruning strategy in DrNAS, maintaining full architectural fidelity during both search and retraining.

- **Dataset**: To prevent overfitting, we partition CIFAR-10 (Krizhevsky, 2009) into two equal subsets: 50% for architecture search (further split 1:1 into training and validation sets) and 50% reserved for post-search architecture retraining.

- **Implementation Details**: Batch sizes, learning rates, and search space configurations are detailed in Table 14. Supernet weights are optimized via SGD (momentum 0.9, weight decay $3 \times 10^{-4}$) with cosine annealing. Architecture parameters use Adam (Kingma and Ba, 2015) ($\beta_1 = 0.5$, $\beta_2 = 0.999$, weight decay $10^{-3}$). All experiments use same 3 initialization seeds (0, 1, 2) for reproducibility.

| Architecture Sampler | Batch Size | | | Learning rate |
|---|---|---|---|---|
| | DARTS-Deep | DARTS-Wide | DARTS-Single-Cell | |
| DARTS | 64 | 96 | 96 | $3 \times 10^{-3}$ |
| DrNAS | 64 | 96 | 96 | $6 \times 10^{-3}$ |
| GDAS | 320 | 480 | 480 | $3 \times 10^{-3}$ |

Table 14: Learning rates and Batch Size used during the search.

## B.2 Architecture Retraining

Following architecture search, we conduct retraining and test using the reserved 50% of CIFAR-10 data (unseen during search) and the standard CIFAR-10 test set, respectively. To ensure robust performance estimation, we train each discovered architecture across nine distinct hyperparameter configurations (denoted HP1–HP9 in Table 15), varying learning rate, and weight decay for a single seed (0). For all the experiments, we retain a batch size of 512. Since we are using only 50% of the dataset compared to DARTS, training requires fewer GPU hours. Thus, all configurations employ SGD optimization with momentum 0.9 for 300 epochs (unlike 600 in DARTS) where weight decay values are configuration-specific as detailed in Table 15. It should also be noted that, unlike DARTS, we use the same fidelity (number of channels, layers) for the retraining as the search.

| Hyperparameter Set | Learning rate | Weight Decay |
|---|---|---|
| HP1 | 0.025 | $1 \times 10^{-4}$ |
| HP2 | 0.025 | $3 \times 10^{-4}$ |
| HP3 | 0.025 | $1 \times 10^{-3}$ |
| HP4 | 0.1 | $1 \times 10^{-4}$ |
| HP5 | 0.1 | $3 \times 10^{-4}$ |
| HP6 | 0.1 | $1 \times 10^{-3}$ |
| HP7 | 0.01 | $1 \times 10^{-4}$ |
| HP8 | 0.01 | $3 \times 10^{-4}$ |
| HP9 | 0.01 | $1 \times 10^{-3}$ |

Table 15: Hyperparameter set used during the retraining of architectures.

## B.3 Hardware Specification

All the experiments use NVIDIA GeForce RTX 2080 Ti GPUs (11GB Memory). Each experiment run consumed eight CPU cores of AMD EPYC 7502 32-Core Processor. For search, the total compute taken per benchmark group is given in Table 16. For retraining architecture, we train around 63 architectures for each of the 9 hyperparameter sets defined in 15, which in total consumes around $\sim 1320$ GPU hours.

| Benchmark | Time (GPU Hours) |
|---|---|
| Deep + All Skip | 290.57 |
| Deep + No skip | 283.44 |
| Deep + Regular | 293.11 |
| Wide + All skip | 69.01 |
| Wide + No skip | 60.28 |
| Wide + Regular | 64.02 |
| Single Cell + All Skip | 70.93 |
| Single Cell + No Skip | 73.42 |
| Single Cell + Regular | 63.21 |

Table 16: Total time per benchmark, aggregated across all methods and their respective three seeds.

## C  Advanced Code Example

Listing 2 presents an advanced example of executing a search within the DARTS search space using the DrNAS sampler. The DrNAS profile comes pre-configured with the core components of the DrNAS method; however, users are not limited to these default configurations. The framework provides flexibility to experiment with various techniques, such as incorporating Operation-Level Early Stopping (OLES), leveraging partial connections, applying SmoothDARTS-based random or adversarial perturbations, entangling convolutional operation weights, utilizing LoRA-based convolutional modules (Krishnakumar et al., 2024), and pruning the network at specific epochs.

For each experiment, the library systematically logs key artifacts, including genotypes, model checkpoints, and standard output logs, in a structured local directory. Additionally, it offers seamless integration with Weights & Biases (Biewald, 2020) for comprehensive experiment tracking. Beyond standard training and validation metrics, the library records a rich set of auxiliary statistics that provide deeper insights into the search process. These include gradient matching scores between validation and training gradients of operations, temporal changes in gradient norm statistics for cells and architectural parameters, the frequency of operations selected as architectural parameters evolve, and the trajectory of architecture values across epochs. Such detailed logging facilitates a deeper understanding of the architecture search dynamics and aids in diagnosing and refining search strategies.

```python
from confopt.profile import (
    DARTSProfile,
    GDASProfile,
    ReinMaxProfile,
    DrNASProfile,
)
from confopt.train import Experiment
from confopt.enums import DatasetType, SearchSpaceType

if __name__ == "__main__":
    search_space = SearchSpaceType.DARTS

    # use any profile from
    # DARTSProfile, GDASProfile, ReinMaxProfile, DrNASProfile
    profile = DrNASProfile(
        trainer_config=search_space,
        epochs=50,
        perturbation="random",  # sdarts options- "random"/ "adversarial"
        entangle_op_weights=True, # option for weight entanglement
```

```python
    lora_rank=1, # option for LoRA
    lora_warm_epochs=5,
    prune_epochs=[3, 6], # Pruning configurations
    prune_fractions=[0.2, 0.2],
    oles=True, # Use OLES
    calc_gm_score=True,
    is_partial_connection=True, # Use partial channels
)

# configure the searchspace paraemeters
profile.configure_searchspace(layers=4, C=76)

profile.configure_lora(
    lora_alpha=2.0, # lora scaling factor
)

profile.configure_perturbator(
    epsilon=0.1,  # epsilon pertubation to add to arch parameters
)

# Configure the Trainer
profile.configure_trainer(
    lr=0.03,  # lr of model optimizer
    arch_lr=3e-4,  # lr of arch optimizer
    optim="sgd",  # model optimizer
    arch_optim="adam",  # arch optimizer
    optim_config={  # configuration of the model optimizer
        "momentum": 0.9,
        "nesterov": False,
        "weight_decay": 3e-4,
    },
    arch_optim_config={  # configuration of the arch optimizer
        "weight_decay": 1e-3,
    },
    scheduler="cosine_annealing_lr",
    batch_size=4,
    train_portion=0.7,  # portion of data to use for training the model
    checkpointing_freq=5,  # How frequently to save the supernet
)

# Add any additional configurations to this run
# Used to tell runs apart in WandB, if required
profile.configure_extra(
        # Name of the Wandb Project
        project_name= "my-wandb-projectname",
        # Purpose of the run
        run_purpose= "my-run-purpose",
)

experiment = Experiment(
    search_space=search_space,
    dataset=DatasetType.CIFAR10,
    seed=9001,
    log_with_wandb=True,  # enable logging with Weights and Biases
```

```
    )

    experiment.train_supernet(profile)
```

Listing 2: Example of advanced code to run GDAS on the DARTS search space

