# OpenReview forum: "$\texttt{confopt}$: A Library for Implementation and Evaluation of Gradient-based One-Shot NAS Methods"
_automl.cc/AutoML/2025/ABCD_Track — AutoML 2025 ABCD Track_

### Review · Reproducibility_Reviewer_KdVT · 2025-04-22

**Comments To Authors:**

1. Submission checklist:\
Present and in line with the contents of the paper.

2. Installation of libraries:\
Instructions are present, clear and functional. Readme files indicates well how to get started with the project.

3. Minimal examples:\
Provided and functional. They give a good starting point in the project and have decent run times.

4. Presented results:\
An additional Readme guides the user through the re-production of the results.
We ran the complete workflow (generating genotypes and training model) for one of the nine evaluations presented.
The authors also provided their computed genotypes which allows the user to easily reproduce more results with a significantly lower computational cost. Which given the cost of the experiments is helpful. The experiment's results reproduced are in line with the metrics presented in the paper.

5. Code:\
The code is clear and easily understandable. There could have been more comments on main classes like Experiment.

6. Conclusion:\
This project provides everything the user needs to replicate the results presented and use this project for their research wether it is for benchmarking new optimisers, setups or datasets.

**Review Confidence:**

4

**Review Rating:**

9

---

### Official Review · Reviewer_PZTW · 2025-04-30

**Comments To Authors:**

Factual aspects:
- State/summarize the main contributions of the paper in a few sentences.
Within this paper, the authors introduce a library for benchmarking gradient-based one-shot NAS methods in a robust and comprehensive manner. The paper also introduces a DARTS benchmark in order to provide a more comprehensive comparison for other gradient-based NAS methods, or NAS methods in general, as stated. Finally, the paper strongly evaluates the necessity for the novel benchmark and complements this with some results showcasing the varying performances of 7 different NAS techniques accross 9 different evaluations.

- Compare the paper with previous work. In particular, is there highly relevant previously published work that the authors do not seem to be aware of?
This work does a great job highlighting previous literature within the field and analyzing the need for a better, albeit somewhat niche, benchmark.

- Express your level of confidence in the correctness of the results, and point out any major errors, if any are found.
I am strongly confident in the findings of this paper.

- Final assessment:
- What are the strengths of the paper? (results? new research direction? application? etc.)
This paper includes a strong evaluation on the faults of current shortcomings within the comparisons between varying gradient-based one-shot NAS, and modified DARTS algorithms, or just using DARTS against other NAS algorithms. This is strongly useful within the field at the moment, because of the absolute prevalence of DARTS as a baseline performance benchmark against other NAS algorithms. This paper also includes a strong experimental design and experimental details within the appendix.


- What are the weaknesses of the paper?
One weakness of the paper is the limited scope of the benchmark strictly to gradient-based one-shot NAS algorithms.

- Express and explain your opinion regarding whether the contributions of the paper (assuming they are correct and original) are interesting/useful/relevant.
This paper strongly positions confopt as a novel and useful library for anyone researching gradient-based one-shot NAS.

- Final Recommendation: Give a final recommendation for acceptance/rejection (or a more refined distinction, such as borderline).
Accept. This paper will benefit the community with its acceptance to the conference.

- Additional feedback: Comment on the quality, clarity, and readability of the writing. Provide comments that may help the authors in producing a revised version of the paper.
This paper is very well written, and provides a great background on the problem faced with current evaluations against DARTS.

**Review Confidence:**

4

**Review Rating:**

9

---

### Official Review · Reviewer_yu1C · 2025-04-30

**Comments To Authors:**

The paper is not double blind, I am not sure if this is a problem.

The authors bring up a valid problem with one-shot NAS benchmarks, reliance on DARTS and disparate repositories. A minimal API to integrate new search spaces is very, very important to standardize NAS research. Reported results do indeed depend heavily on the seeds, making comparison difficult.

The paper motivation is good, explanation of the background on DARTS and bi-level optimization is solid. Their DARTS-Bench-Suite addresses the challenges discussed. Their study on hyper-parameter importance + Figure 3 is insightful, paired with Figure 4 (Apx A) on rank disparity, it will motivate better design of base experiments.

Weaknesses: None, it is a solid, well written paper backed up a good framework. Double blind may be an issue, unsure.

Strengths: Good design, well motivated, useful for the AutoML Community. Very relevant contribution and a good step towards standardizing gradient-based NAS.

**Review Confidence:**

5

**Review Rating:**

9

---

### Official Review · Reviewer_Cqtv · 2025-05-04

**Comments To Authors:**

**Summary**
This paper is motivated by current problems with gradient-based single-shot NAS evaluation: the reliance/saturation of performance on the DARTS search space, and bespoke code repositories for every new technique. To address these issues, the authors propose a library called "configurable optimizer" or "confopt," which makes it easier to develop and benchmark gradient-based one-shot NAS methods, and propose a suite of variants of the DARTS search space, called DARTS-Bench-Suite. The authors use these new tools to evaluate existing gradient-based single-shot NAS methods and find substantial variation in their performance across the search spaces in DARTS-Bench-Suite.

**Strengths**
- The authors tackle an important problem and take a systematic approach to breaking down the central issues with gradient-based one-shot NAS eval. I am particularly intrigued by the insight that there is a poor rank correlation between the proxy supernet search space and the downstream discretized target space. I also think that the point about hyperparameter being a confound is an important one that the authors mention.
- The proposed API for confopt seems incredibly useful, streamlined, and seem to pick out the right parts of the NAS development pipeline that users would want to vary, while maintaining flexibility. It is helpful that the authors have reimplemented several existing methods using confopt.
- Normally I would comment on the usage of only one dataset -- CIFAR -- but in this case it seems warranted, and in fact a smart design decision for this particular paper since there are so many confounds and complexities even when we evaluate on a single dataset. That said, I am wholly in favor of the statement at the end of the paper that "More generalizable findings could be obtained by introducing more diverse, real-world datasets," as potential followup work, once the community converges on a robust evaluation pipeline (this work clearly takes steps towards this goal).

**Weaknesses**
- This is not necessarily a weakness, but do the authors have experimental results showing the poor rank correlation between the proxy supernet and the target space? This would be a useful motivating result that would make the narrative even more compelling, even if the experiment is relatively simple (e.g. showing a poor rank correlation across, say, 10 high-performing DARTS proxy and target architectures).
- The choices of DARTS variants in the proposed DARTS-Bench-Suite seem reasonable, but it would also be useful to see for instance, how rankings between methods might gradually change if one were to run all of them across an interpolation between two of the supernets. For instance, the authors provide DARTS-Wide and DARTS-Deep, but one could also construct several intermediate supernets that trade-off width and depth. My point is this: I would hope to see some kind of gradual change in rankings across this sweep -- a major risk here is that the rankings change quite chaotically as you gradually change the supernet, which could suggest that even with the proposed improvements to the NAS evaluation pipeline, evaluation might still be too random. This seems important to me, because it feels like this risk could shift the narrative of the paper. If the change in ranking is indeed more gradual, then this is useful information, and could clue us in to what aspects of search spaces different approaches tend to struggle with.
- The authors evaluate whether the choice of hyperparameters impacts the final ranking of NAS methods, and find that it does. This is a great question to address, but I am curious how these 9 hyperparameter configs were chosen. Are they random? I feel that this part can be clarified, somewhat, as it seems to be an important point.

**Review Confidence:**

4

**Review Rating:**

8

---

### Meta-Review · Area_Chair_nCxw · 2025-05-04

**Recommendation:** Accept
**Confidence:** 5

**Metareview:**

Well written paper, all agreed to accept.